# Eupatilin Improves Cilia Defects in Human CEP290 Ciliopathy Models

**DOI:** 10.3390/cells12121575

**Published:** 2023-06-07

**Authors:** Julio C. Corral-Serrano, Paul E. Sladen, Daniele Ottaviani, Olivia F. Rezek, Dimitra Athanasiou, Katarina Jovanovic, Jacqueline van der Spuy, Brian C. Mansfield, Michael E. Cheetham

**Affiliations:** 1UCL Institute of Ophthalmology, 11-43 Bath Street, London EC1V 9EL, UK; p.sladen.16@alumni.ucl.ac.uk (P.E.S.); daniele.ottaviani@unipd.it (D.O.);; 2Department of Biology, University of Padova, Padova, 35122 Padova PD, Italy; 3Eunice Kennedy Shriver National Institute of Child Health and Human Development, National Institutes of Health, 6710B, Rockledge Drive, Montgomery County, MD 20892, USA

**Keywords:** CEP290, Leber congenital amaurosis (LCA), retina, ciliopathy, photoreceptor, cilium, eupatilin, retinal organoid

## Abstract

The photoreceptor outer segment is a highly specialized primary cilium that is essential for phototransduction and vision. Biallelic pathogenic variants in the cilia-associated gene *CEP290* cause non-syndromic Leber congenital amaurosis 10 (LCA10) and syndromic diseases, where the retina is also affected. While RNA antisense oligonucleotides and gene editing are potential treatment options for the common deep intronic variant c.2991+1655A>G in *CEP290*, there is a need for variant-independent approaches that could be applied to a broader spectrum of ciliopathies. Here, we generated several distinct human models of *CEP290*-related retinal disease and investigated the effects of the flavonoid eupatilin as a potential treatment. Eupatilin improved cilium formation and length in CEP290 LCA10 patient-derived fibroblasts, in gene-edited *CEP290* knockout (CEP290 KO) RPE1 cells, and in both CEP290 LCA10 and CEP290 KO iPSCs-derived retinal organoids. Furthermore, eupatilin reduced rhodopsin retention in the outer nuclear layer of CEP290 LCA10 retinal organoids. Eupatilin altered gene transcription in retinal organoids by modulating the expression of rhodopsin and by targeting cilia and synaptic plasticity pathways. This work sheds light on the mechanism of action of eupatilin and supports its potential as a variant-independent approach for *CEP290*-associated ciliopathies.

## 1. Introduction

Primary cilia are microtubule-based organelles that function as a sensor of the extracellular environment. Photoreceptor cells contain a highly specialized primary cilium called the outer segment that is essential for detecting light and initiating phototransduction. Diseases that affect the function of cilia genes are termed ciliopathies, many of which include retinal degeneration as a common feature. Biallelic variants in the cilia gene *CEP290 (NPHP6)* can cause a series of syndromic ciliopathies, including Joubert syndrome, nephronophthisis, Meckel–Gruber syndrome and Senior–Loken syndrome [1,2,3]. *CEP290* is also the most affected gene in non-syndromic Leber congenital amaurosis (type 10, LCA10), which is characterized by early vision loss due to photoreceptor cell death [4,5]. The most common *CEP290* deep intronic variant, c.2991+1655A>G (over 100 bp away from the nearest exon–intron boundary), results in aberrant splicing and the inclusion of a cryptic exon containing an immediate premature stop codon, p.Cys998Ter [4]. 

Current therapeutic options to treat LCA10 are limited and have focused on this deep intronic variant [6]. For example, a gene-editing approach directed to this intronic variant was in clinical trials, but these have been paused [7]. Moreover, treatment of the *CEP290* c.2991+1655A>G variant with the RNA antisense oligonucleotide (AON) sepofarsen (QR-110) restored CEP290 function in preclinical models [8], improved visual acuity in phase Ib/II clinical trials [9,10,11], and improved cone sensitivity in individuals with congenital blindness [12]. The effects were long lasting, with improvements reported over 12 months following a single injection [10]. However, in a placebo-controlled phase II/III clinical trial, sepofarsen did not meet the primary endpoint of best-corrected visual acuity at month 12, resulting in the discontinuation of the clinical trials [13]. Considering that other *CEP290* pathogenic variants contributing to LCA10 or other syndromic diseases may not respond to AONs or variant specific gene editing, it is important to investigate the potential of novel approaches that have a broader range of action.

Flavonoids are plant-derived small molecules that can influence synaptic plasticity and potentially exert cognitive benefits [14,15,16]. A phenotype-based screen of compounds that could restore ciliation and cilium length in CEP290^null^ RPE1 cells identified the flavonoid eupatilin (5,7-dihydroxy-3’,4’,6-trimethoxyflavone) as a candidate therapeutic molecule for retinal ciliopathies [17]. Treatment of the rd16 *Cep290* mouse model of retinal dystrophy showed increased photopic responses and improved M-opsin immunofluorescence in the outer retina, although no effect on rhodopsin traffic or rod cell function were reported [17]. Eupatilin was also reported to rescue ciliary gating in the absence of Rpgrip1l (nphp8) in mouse embryonic fibroblasts [18]. Both CEP290 and RPGRIP1L are parts of the ciliary transition zone (TZ), a cilia domain distal to the basal body that functions as a gatekeeper of cilia protein import and export [19,20,21]. Eupatilin has previously been used in a clinic as a drug treatment for gastritis and peptic ulcer, with reportedly limited adverse effects [22].

In this study, we investigated the effect of eupatilin in different models of human *CEP290* disease, including patient-derived fibroblasts, CRISPR-induced RPE1 knockout cells, and induced pluripotent stem cell (iPSC)-derived retinal organoids, evaluating its potential as a variant-independent therapeutic approach for the treatment of ciliopathies.

## 2. Materials and Methods

### 2.1. Cell Lines

The hTert-RPE1 control cells were a kind gift from Ronald Roepman (Radboud University Nijmegen, The Netherlands). Guide RNAs (gRNAs) were designed to target CEP290 exon 6 and cloned into the pSpCas9(BB)-2A-GFP (PX458; a gift from Feng Zhang; Addgene plasmid #48138). RPE1 cells were transfected with this plasmid by using Lipofectamine 3000 (Invitrogen, Carlsbad, CA, USA). Transfected cells were subjected to puromycin selection after transfection and subsequently seeded as single cells into 96-multiwell plates in order to screen for CEP290 KO clones. 

CEP290 LCA10 patient fibroblasts were obtained from a male individual, following informed consent as described [23]. The control iPSCs used in this work were reprogrammed from healthy human dermal fibroblasts of neonatal origin (HDFn) and were characterized previously [23,24]. 

Isogenic CEP290 KO stem cells were generated using a simultaneous cellular reprogramming and CRISPR/Cas9 gene editing protocol [25,26], by targeting CEP290 exon 6 in HDFn iPSCs. Control HDFn cells were nucleofected using the Cell Line Nucleofector Kit R (Lonza) containing 1μg of each episomal reprogramming vector [27] and 1 μg of the CEP290-targeting PX458 plasmid.

Following nucleofection, cultures were maintained until the presence of iPSC colonies emerged. Individual clones were mechanically isolated and placed in individual wells of a geltrex-coated 12-well plate. Clonal iPSC lines were subsequently amplified, before DNA extraction, using the Wizard SV genomic DNA extraction kit (Promega, Madison, WI, USA) and following the manufacturer’s instructions. The CRISPR/Cas9 target region was expanded using standard PCR and analyzed by Sanger sequencing to detect the presence or absence of CRISPR/Cas9-induced variants by aligning the sample sequence data to the CEP290 reference sequence (ENST00000552810, ensembl.org) on Benchling (Benchling.com). 

### 2.2. qPCR

Total RNA was extracted using the RNeasy Mini Kit (QIAGEN, Hilden, North Rhine Westphalia, Germany) following the manufacturer’s instructions. First, strand cDNA synthesis was performed using the Tetro cDNA synthesis kit (Bioline Reagents, London, UK). Quantitative PCR (qPCR) was completed using the SYBR green method, carried out on a QuantStudio 6 Flex Real-Time PCR System (Applied Biosystems, Carlsbad, CA, USA) using LabTaq Green Hi Rox (Labtech, Heathfield, East Sussex, UK). Primers are listed in Appendix A. Relative gene expression levels were determined using the ΔΔCt method, with the geometric mean of *GAPDH* and *ACTIN* as the reference. GraphPad Prism v.8 (GraphPad Software, Inc.) was used for statistical analyses and for generating plots. 

### 2.3. Western Blot

Total protein samples were lysed in radioimmunoprecipitation (RIPA) buffer containing a 2% protease inhibitor cocktail (Merck Life Science UK Limited, Sigma, Gillingham, Dorset, UK). Protein quantification was completed using the Pierce Bicinchoninic acid (BCA) protein assay (Life Technologies Europe BV, ThermoFisher, Bleiswijk, The Netherlands), following the manufacturer’s microplate procedure. Equal amounts of protein were run on 8% acrylamide gels before protein transfer to nitrocellulose membranes via wet transfer. Protein membranes were incubated with primary antibodies for CEP290 or relevant reference proteins overnight at 4 °C before incubation with HRP-conjugated secondary antibodies. Finally, proteins were detected using Clarity Western ECL substrate (BioRad) and imaged with ImageLab on a BioRad ChemiDoc XRS+. Quantification of Western blots was performed with ImageLab software (bio-rad.com) and target protein expression was normalized to reference proteins before analysis.

### 2.4. Cell Culture and Eupatilin Treatment

hTERT-RPE1, fibroblasts, or iPSC were grown on chamber slides until 90% confluency was reached. To induce cilia growth, fibroblasts and hTERT-RPE1 cells were incubated with serum starvation medium (DMEM/F12, 0.2% FBS + 1% Pen/Strep + 1% Sodium pyruvate), while iPSC were cultured as normal with mTESR Plus medium. 

Fresh eupatilin (Adipogen; AG-CN2-0432-M025) was added to the cells at the corresponding concentration for 24 h before immunostaining.

### 2.5. Immunocytochemistry

Cultured cells were fixed in 2% paraformaldehyde (PFA) for 15 min at room temperature (RT), followed by 1% Triton-X-100 treatment for 5 min and blocking in 2% fetal bovine serum (FBS) for 20 min. Subsequently, cells were incubated with primary antibodies diluted in blocking solution for 1 h. After incubation, cells were washed three times in phosphate buffered saline (PBS) and incubated with the corresponding Alexa Fluor conjugated secondary antibody (ThermoFisher). After secondary antibody incubation, DAPI (ThermoFisher) was incubated for 5 min. Finally, slides were washed three times in PBS for 5 min and mounted in DAKO fluorescence mounting medium (Agilent, Santa Clara, CA, USA).

### 2.6. iPSC Differentiation to Retinal Organoids

iPSCs were differentiated to retinal organoids, as previously described, with small variations [28]. Briefly, the cells were seeded on Geltrex-coated (Thermo Fisher) six-well plates with mTESR Plus Medium (Stem Cell Technologies, Vancouver, Canada) until 90–95% confluency. Essential 6™ Medium (Thermo Fisher) was added to the culture for 2 days, followed by the addition of neural induction media (Advanced DMEM/F-12 (1:1), 1% N2 supplement, 2 mM GlutaMax, and 1% Pen/Strep) until neuro retinal vesicles (NRVs) appeared. On day 6 of differentiation, a single treatment with 1.5 nM BMP4 (Prepotech, Cranbury, NJ, USA) was added, and half-media changes were carried out until day 16. NRVs were excised and kept in ultra-low binding 96-well plates for maturation. For retinal differentiation and maturation, serum-free retinal differentiation media was added (DMEM/F12 (3:1), 2% B27, 1% non-essential amino-acids, and 1% Pen/Strep) for 6 days after collection. For retinal maturation, media was supplemented with 10% FBS, 100 µM Taurine, and 2 mM GlutaMax. Retinoic acid (1 µM) was added on day 50. On day 70, N2 was added to the media (RMM2), and the concentration of retinoic acid was reduced to 0.5 µM. To promote photoreceptor differentiation, retinoic acid was removed from the media on day 100.

### 2.7. Cryopreservation and Immunohistochemistry of Retinal Organoids 

Retinal organoids were briefly washed once in PBS and placed in a mixture of 4% PFA + 5% sucrose in PBS for 40 min. Organoids were then placed in 6.25% sucrose in PBS for 1 h, followed by 12.5% sucrose in PBS for 30 min and in 25% sucrose in PBS for 1 h. All incubation steps were performed at 4 °C. Organoids were then embedded in OCT, slowly frozen on dry ice, and stored at −80 °C until cryosectioning. Cryosections (7 µm thick) were collected and stored at −20 °C for later analysis.

For immunohistochemistry, sections were first washed once in PBS and blocked with 10% donkey serum, 0.05% Triton X-100 in PBS for 30 min. The primary antibodies used were the following: anti-rabbit CEP290 (Abcam, Cambridge, UK) ab84870, 1:100; anti-rabbit ARL13B (Proteintech, Manchester, UK) 1:1000; anti-mouse Pericentrin (Abcam) ab28144, 1:1000; anti-mouse Rhodopsin 4D2 (Millipore) MABN15 1:1000; anti-rabbit Opsin Antibody, Red/Green (Millipore) 1:500; anti-mouse CRX (Abnova) H00001406-M02 1:1000; and anti-rabbit Recoverin (Millipore) AB5585 1:1000. The primary antibodies were incubated in the blocking solution diluted 50% in PBS for 1 h. Sections were then washed in PBS and incubated with Alexa Fluor (Thermo Fisher) secondary antibodies in the diluted blocking solution for 45 min. Nuclei were visualized using DAPI (2 μg/mL). Samples were washed in PBS and mounted using DAKO fluorescence mounting medium (Agilent).

### 2.8. Cilia Quantification and Statistical Analysis

Images were obtained using a Carl Zeiss LSM700 laser-scanning confocal microscope or a Leica Stellaris 8 confocal microscope. At least 3 images were taken for each condition. For RPE1 and fibroblasts, cilia were segmented and cilium length was quantified using the plugin CiliaQ on Fiji/ImageJ based on the ARL13B channel (and basal body present in the PCN channel) [29]. Manual counting was performed to measure photoreceptor ciliation and cilia length in retinal organoids. Statistical analyses were carried out using Prism and are indicated at the bottom of each figure’s legend.

### 2.9. RNAseq Pre-Processing and Analysis 

Total RNA was extracted with the RNeasy micro kit (QIAGEN, Hilden, North Rhine Westphalia, Germany), following the manufacturer’s instructions. Illumina RNA library preparation was performed by Genewiz (Takeley, Essex, UK).

The Illumina NovaSeq platform was used to obtain raw sequencing data, which was subjected to quality control using FastQC. Trimmomatic was used to remove any low-quality sequences, adapter sequences, and other contaminants. Trimmed reads were aligned to the GRCh38 reference genome using the STAR aligner. Gene expression was quantified using the featureCounts program. 

The gene-level count data was imported into RStudio for differential gene expression analysis using the DESeq2 package [30]. The gene counts were normalized using the DESeq2 normalization method, which accounts for differences in library size and gene length. A generalized linear model was fitted to the normalized counts using the negative binominal distribution. The differential gene expression analysis was carried out using the DESeq2 function with apeglm shrinkage [31]. Genes with an FDR-adjusted *p* value cutoff of 0.1 were considered significant. An over-representation analysis was carried out using the clusterProfiler package for identification of cellular components enriched in the significantly differentially expressed gene set. 

## 3. Results

### 3.1. Eupatilin Increases Cilia Incidence and Length in CEP290 LCA10 Patient Fibroblasts

Dermal fibroblasts derived from an affected male individual with LCA10 and homozygous for the *CEP290* c.2991+1655A>G variant were compared to fibroblasts from a control individual (Figure 1). Ciliation was induced and evaluated by immunocytochemistry (Figure 1A). CEP290 LCA10 fibroblasts displayed reduced cilia incidence (Figure 1B) and the cilia present were reduced in length (Figure 1C), confirming previously reported results [23]. Importantly, treatment with 20 µM eupatilin significantly increased cilia incidence (Figure 1B) and rescued cilia length in CEP290 LCA10 patient fibroblasts (Figure 1C), whilst having no significant effect in controls.

### 3.2. Eupatilin Restores Cilia Incidence and Length in CEP290 Knockout RPE1 Cells

In order to validate the reported effects of eupatilin on RPE1 cilia, we made an independent *CEP290* knockout (KO) cell line using CRISPR/Cas9 gene editing (Figure 2). A guide RNA (gRNA) in exon 6 was used to direct Cas9 to cleave *CEP290*. RPE1 clones were selected and screened for insertions or deletions (InDels), resulting in frame shifts. An RPE1 cell line was selected with a homozygous 1 bp deletion (c.C311del; p.Ser104Leufs*20) that led to a frame shift (at amino acid 104) and a premature stop codon after 20 amino acids, similar to a pathogenic variant reported in Joubert and Meckel–Gruber syndromes (c.338T>A: p.Leu113*), and as such it was predicted to be a severe “null” mutation (Figure 2A). Characterization of this cell line showed that the expression of *CEP290* transcript was reduced by 80%, presumably due to nonsense-mediated decay (Figure 2B), and protein expression was reduced to 20% of the isogenic control level (Figure 2C,D). Exon 6 is in-frame with exon 5 and exon 7, leading to the possibility that skipping of exon 6 could produce a truncated protein lacking the 48 amino acids of exon 6 and giving rise to the band detected by immunoblotting (Appendix A). It is not known if this truncated form of CEP290 is functional; however, no CEP290 protein could be detected at the basal body of the CEP290 KO cells by immunocytochemistry, in contrast to the isogenic control RPE1 cells (Appendix A). 

The *CEP290* KO RPE1 cells showed significant reductions in both cilia incidence and cilia length (Figure 2E–G), i.e., fewer and shorter cilia. Importantly, treatment with 20 µM eupatilin led to a significant increase in cilia incidence and length, returning to levels similar to that of the isogenic control cell line (Figure 2E–G). Interestingly, eupatilin treatment also led to a small but significant increase in cilia length in the control RPE1 cells. 

A more detailed investigation of the effect of eupatilin on cilia incidence and length in CEP290 KO RPE1 cells revealed that there was a dose response for the effect of eupatilin on cilia length and incidence (Figure 2H,I); 5 µM eupatilin resulted in a small increase in cilia incidence and length, which did not reach statistical significance. By contrast, both 10 µM and 20 µM eupatilin led to statistically significant increases in cilia incidence and length, with 20 µM eupatilin restoring incidence and length to isogenic control levels (dashed red line). 

### 3.3. Eupatilin Treatment Restores Ciliation and Cilia Length in iPSC-Derived CEP290 LCA10 and CEP290 KO Retinal Organoids

CEP290 KO iPSC were developed in a control line by CRISPR/Cas9 technology and the same guide RNA was used for the CEP290 KO RPE1 cell line. An iPSC line with compound heterozygous single base pair deletions (c.C315del and c.T316del alleles; p.Ser104Leufs*20) was selected (Appendix A). This iPSC line was predicted to cause the same frame shift and premature stop codon as the CEP290 KO RPE1 cells. CEP290 immunoreactivity was undetectable at the cilia of these iPSCs, in contrast to the control iPSC, comparable with the CEP290 KO RPE1 cells (Appendix A). CEP290 LCA10 fibroblasts reprogrammed to iPSC [8,23] were differentiated to retinal organoids, together with the CEP290 KO iPSC line, as previously described [28]. The CEP290 LCA10 and CEP290 KO iPSC produced retinal organoids with a characteristic neuroepithelial morphology that was similar to that of control organoids (CEP290 WT) early in development (Figure 3A). Photoreceptor differentiation was confirmed through cone-rod homeobox (CRX) and recoverin (RCVRN) immunoreactivity in the outer nuclear layer after 120 days of differentiation (D120) (Figure 3B). CEP290 staining confirmed the absence of CEP290 protein in both CEP290 KO and CEP290 LCA10 retinal organoids (Appendix A). Treatment with eupatilin was initiated at 90 days of differentiation (D90), after polarized organization of photoreceptors had occurred, and continued for 30 days before harvesting. 

Cilia morphology was analysed by immunohistochemistry (IHC) at D120. CEP290 LCA10 retinal organoids showed reduced cilia incidence and cilia length, as previously reported [8,23,32]. The CEP290 KO retinal organoids also showed reduced cilia incidence and length, and were comparable with the CEP290 LCA10 organoids (Figure 3C–E). Importantly, treatment with eupatilin at 10 or 20 µM led to significant increases in cilia incidence and length in both CEP290 KO and CEP290 LCA10 retinal organoids (Figure 3D,E). 

### 3.4. Opsin Accumulates in the Outer Nuclear Layer of CEP290 LCA10 Retinal Organoids and Is Rescued by Eupatilin

To test the effect of eupatilin on outer segment protein traffic, we compared opsin localization in CEP290 LCA10 and control retinal organoids at D180. Staining for rhodopsin (RHO) and L/M opsin suggested that there was retention of rhodopsin in the outer nuclear layer (ONL) of the CEP290 LCA10 organoids at D180, compared to control organoids (Figure 4A,B). L/M opsin was equally accumulated in the ONL of controls and CEP290 LCA10 organoids (Figure 4A,B). Interestingly, treatment with 20 µM eupatilin for 30 days (D150 to D180) reduced the accumulation of rhodopsin and L/M opsin within the ONL (Figure 4A,B). 

### 3.5. Eupatilin Treatment Reduces the Expression of Rhodopsin

To determine the potential mechanisms related to the decrease in rhodopsin in the ONL, we performed qPCR analyses on mature (D250) control organoids that had been treated with eupatilin for 30 days (Figure 5A). These analyses revealed that eupatilin treatment caused a significant reduction of over 50% in *RHO* expression at 10 µM. The 20 µM treatment also reduced *RHO* transcript but did not reach statistical significance (Figure 5A). The expression changes in other photoreceptor genes were less pronounced, and only *CRX* was significantly reduced at 10 µM. Both *OPN1MW/OPN1LW* and *ARR3* showed a trend for reduced expression, although not significantly, suggesting a potential effect on cones as well as rods (Figure 5A). The effect of eupatilin on *RHO* and other photoreceptor transcript levels appeared to be transient, however, as the levels were restored to control levels when the samples were allowed to recover for 14 days after 10 µM and 20 µM eupatilin treatment (Figure 5A).

### 3.6. Eupatilin Modulates the Expression of Ciliary and Synaptic Pathways 

To assess the effect of eupatilin on global retinal transcription, we performed bulk RNA sequencing (RNAseq) on mature control organoids treated with 20 µM eupatilin for 30 days, from D210 to D240 (Figure 5B–E). Clustering of the samples by significant differential expression showed that the eupatilin-treated samples had similar expression patterns, distinct from those of the vehicle treated samples (Figure 5B). Out of 56,558 mapped genes, 645 genes were differentially expressed, with 461 downregulated genes and 184 upregulated genes identified at false discovery rate (FDR)-adjusted *p*-value cutoff of 0.1 (Figure 5C). Filtered by adjusted *p*-values, the most significantly downregulated genes were *PRR15*, a gene with unknown function but exclusively expressed in post-mitotic cells; *NRXN3*, a presynaptic adhesion molecule that regulates neurotransmitter release [33]; the cone photoreceptor kinase *GRK7* that catalyzes phosphorylation of cone opsins [34]; and *TENM2*, a receptor involved in calcium-mediated signaling in synapses [35]. The most significantly upregulated gene was the heat shock response gene *HSPA6*, which encodes an inducible Hsp70 protein (Figure 5C). Over-representation analysis revealed two major sets of genes transcriptionally affected by eupatilin: one involving photoreceptor neuron synapse genes and one involving cilia genes (Figure 5D). Interestingly, eupatilin altered the expression of essential phototransduction genes involved in several forms of retinal degeneration, including *CRB1*, *EYS*, *SAG*, *GRK7*, *RPGRIP1*, *USH2A*, *PROM1*, *ARR3*, *PDC*, *PDE6A*, *GNGT1*, and *GUCA1C*; the photoreceptor cation channels *CNGA1* and *CNGB1*; and the cilia transmembrane protein genes *PKHD1*, *PKD2L1*, *PCDHB15*, *CDH23* (*USH1D*), and *PCDH15* (*USH1F*) (Figure 5E). *RHO* expression was also reduced, although it did not reach statistical significance (*p*-value = 0.258, not shown).

## 4. Discussion

Leber congenital amaurosis type 10 (LCA10) is a severe retinal disease caused by biallelic variants in the ciliary gene *CEP290*. Despite the recent clinical trials of gene-editing and RNA-targeted therapies [8,9,10,11,12,36,37], effective treatments are still limited. Plant flavonoid supplementation is being studied for the treatment of a range of different ocular disorders [38]. For example, baicalein has been found to lower intraocular pressure in glaucoma [39] by regulating PI3K/AKT signaling [40], while quercetin provides neuroprotective effects in diabetic rat retinas [41] and in mouse models of light-induced retinal degeneration [42].

Eupatilin (5,7-dihydroxy-3’,4’,6-trimethoxyflavone) is a flavone derived from the plant genus *Artemisia* with different anti-oxidant [43,44], anti-ulcerative [45], anti-cancer [46,47], and anti-inflammatory [48,49] properties [50]. Eupatilin is prescribed clinically in South Korea for the treatment of gastritis and peptic ulcers under the name of Stillen^TM^ [51]. Previously, it has been reported that eupatilin can rescue cilia defects due to the absence of CEP290 in RPE1 CEP290^null^ cells [17]. In addition, eupatilin helped to rescue cilia TZ defects in *Rpgrip1l^-/-^* mouse embryonic fibroblasts [18]. CEP290 and RPGRIP1L are TZ proteins that are important for regulating the entry and exit of proteins to cilia [19]. In the retina, CEP290 is required for RPE maturation and functional polarization [52], as well as for photoreceptor ciliogenesis [53]. In RPE and other primary cilia, CEP290 localizes to the TZ, but in photoreceptors, CEP290 localizes along the length of the connecting cilium [54] and follows a 9-fold symmetry close to the Y-links [55]. This supports a role for CEP290 in photoreceptor cilia that are different from other primary cilia. 

The aim of this study was to validate and extend the previous report by Kim and colleagues [17] on the effect of eupatilin as a variant-independent approach to treat CEP290 ciliary disease. We confirmed the beneficial effect on ciliation and cilium length in a newly generated CRISPR/Cas9 CEP290 KO RPE1 cell line. We also demonstrated this beneficial effect in CEP290 LCA10 patient fibroblasts. This effect of eupatilin was dose-dependent, with 10 and 20 µM being most beneficial in RPE1 cells. This dose-dependent effect was reported with ciliation rescue plateauing with concentrations above 20 µM in cells, and 40 mg/kg was sufficient for opsin rescue in the mouse retina [17]. Furthermore, we showed that eupatilin can also rescue photoreceptor cilia defects in CEP290 KO and CEP290 LCA10 iPSC-derived retinal organoids.

We have previously reported that patient-derived CEP290 LCA10 fibroblasts present ciliation and cilium length defects [23]. These defects are caused by the inclusion of a cryptic exon in CEP290 between exons 26 and 27 that causes reduced CEP290 expression (40–50% of control) in fibroblasts and delays in ciliation, whereas there is little (10–20%) correctly spliced CEP290 expression in retinal organoids [23]. Similarly, we showed that CEP290 KO RPE1 cells have reduced CEP290 expression (10–20% of control), but not total ablation, potentially due to partial exon-6-skipping and production of an in-frame truncated protein [56]. We identified a potentially important phenotype in CEP290 LCA10 retinal organoids: the accumulation of visual opsins in the outer nuclear layer of photoreceptors, which is reduced by eupatilin. We also showed that eupatilin modulates gene transcription to reduce *RHO* expression. This reduction of *RHO* expression might be responsible for the observed reduced rhodopsin ONL accumulation. Thus, reducing the burden of cilia traffic in a ciliopathy could be beneficial and might also assist the transport of the remaining protein. Our observations contrast with previous work in mice by Kim and colleagues, where they observed a rescue of cone opsins in the outer segments, but no changes in rhodopsin. Differences between murine and human models may play a role in this discrepancy, especially taking in consideration the differences in photoreceptor composition and splicing mechanisms between humans and mice [57]. 

Eupatilin was suggested to exert its effect on CEP290 ciliopathy at the protein level through calmodulin binding and release of NPHP5 to the transition zone [17]. In NPHP5-LCA retinal organoids, CEP290 expression is reduced, and rhodopsin mislocalizes to the ONL. In contrast to CEP290, NPHP5 LCA patient fibroblasts and iPSC-RPE present abnormally long cilia [58], similar to what was reported in RPGRIP1L KO cells [18]. This could be explained by aberrant cargo transport of ciliary proteins due to reduced CEP290. The depletion of CEP290 affects the cilium in a different way than the absence of NPHP5 or RPGRIP1L, since it interferes with the ciliary microtubule organization, thus reducing ciliation and cilium length [59].

Calmodulin is a Ca^2+^ binding protein that interacts with other centrosomal proteins with Ca^2+^ binding properties such as centrin, CEP110 (a CEP290 interactor), or CEP97 [60,61,62,63]. In photoreceptors, calmodulin acts by regulating calcium storage in phototransduction [64]. Eupatilin could be acting, therefore, at different levels in the photoreceptor cell, in the outer segments, and at the cilia base. We did not observe changes in the expression levels of *CALM1*, which encodes calmodulin, in the RNAseq data; however, some Ca^2+^-related genes similar to calmodulin, such as calcium-binding protein 1 (*CABP1*), *CABP4*, *CAMK4*, *CADPS*, and *CALB1*, were downregulated. The calcium-binding protein DREAM, from the recoverin family, functions as a calcium-regulated transcriptional repressor [65,66] and its modulation could have therapeutic potential for neurological diseases such as Alzheimer’s disease [67]. This suggests that targeting calcium signaling pathways with compounds such as eupatilin could be useful for modulating other neurodegenerative diseases.

RNAseq analysis also revealed that eupatilin downregulated two major gene modules: one module involving neural synapse genes (e.g., *TENM2*, *NRXN3*) and a second module involving photoreceptor cilia genes (e.g., *EYS*, *RPGRIP1*, *USH2A*, *PROM1*, *PKHD1*, *ARR3*, *GRK7*, *RRH*, *SAG*, *PDE6A*, and *CNGA1*). The effect of eupatilin at the transcriptional level on phototransduction or outer segment genes has not been previously reported. Studying the interaction of eupatilin with cilia genes in more detail might reveal other potential and more specific targets that could be used to treat CEP290-related disease. Some studies suggest a role of flavonoids in modulating synaptic plasticity [68], but the precise role and the mechanisms by which flavonoids modulate cognitive functions are yet to be fully established [69,70]. 

While most of the transcriptional effects of eupatilin appear to be downregulation, some genes were upregulated by eupatilin. The most significantly upregulated gene was the heat-shock response gene *HSPA6*, while *HSPA12A* was also upregulated, suggesting that eupatilin might activate the heat-shock response. In certain models of retinal degeneration, *HSPA1A* overexpression is thought to play a protective role in stressed photoreceptors [71,72]. For instance, *HSPA6* and *HSPA1A* contribute to the protection of differentiated human neuronal cells from cellular stress [73,74]. However, the role of *HSPA6* and *HSPA12A* in photoreceptors is still not known. Future studies could investigate the transcriptional effect of eupatilin in CEP290 LCA10 retinal organoids to better understand CEP290 retinal disease, while scRNA sequencing could help in the understanding of the impact of eupatilin on the different retinal cell populations.

In summary, we confirmed the beneficial effect of eupatilin on different CEP290 ciliopathy models, including CEP290 retinal organoids. We also provided novel insights into the effect of eupatilin at the transcriptional level in control retinal organoids. Taken together, this work supports eupatilin as a potential variant-independent approach for CEP290-associated ciliopathies.

## Figures and Tables

**Figure 1 cells-12-01575-f001:**
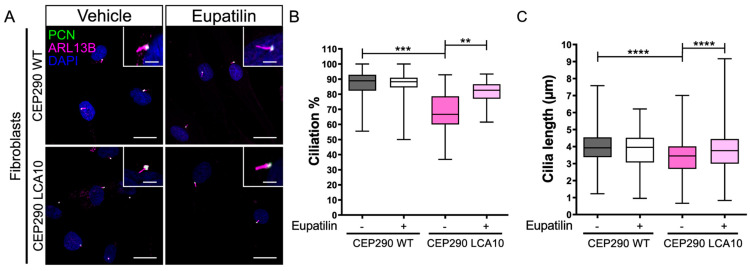
Eupatilin increases cilia incidence and length in CEP290 LCA10 patient fibroblasts. (**A**) Representative images of control (CEP290 WT) and CEP290 LCA10 fibroblasts treated with eupatilin (20 µM) or vehicle, stained for pericentrin (PCN, green, basal body) and ARL13b (magenta, axoneme). Scale bar: 20 µm. Inset shows cilia at higher magnification. Scale bar 5 µm. (**B**,**C**) Cilia incidence and length were scored using the elaboration of an ARL13B positive axoneme from a PCN positive basal body. Number of cilia counted in B: WT vehicle (-) *n* = 89, Eupatilin (+) *n* = 147; CEP290 LCA10 vehicle (-) *n* = 99, Eupatilin (+) *n* = 87 (B). Number of cilia counted in C: WT vehicle (-) *n* = 271, Eupatilin (+) *n* = 231; CEP290 LCA10 vehicle (-) *n* = 214, Eupatilin (+) *n* = 345. Mean ± standard deviation (SD), ** *p* < 0.01, *** *p* < 0.005, **** *p* < 0.0001 one-way ANOVA test.

**Figure 2 cells-12-01575-f002:**
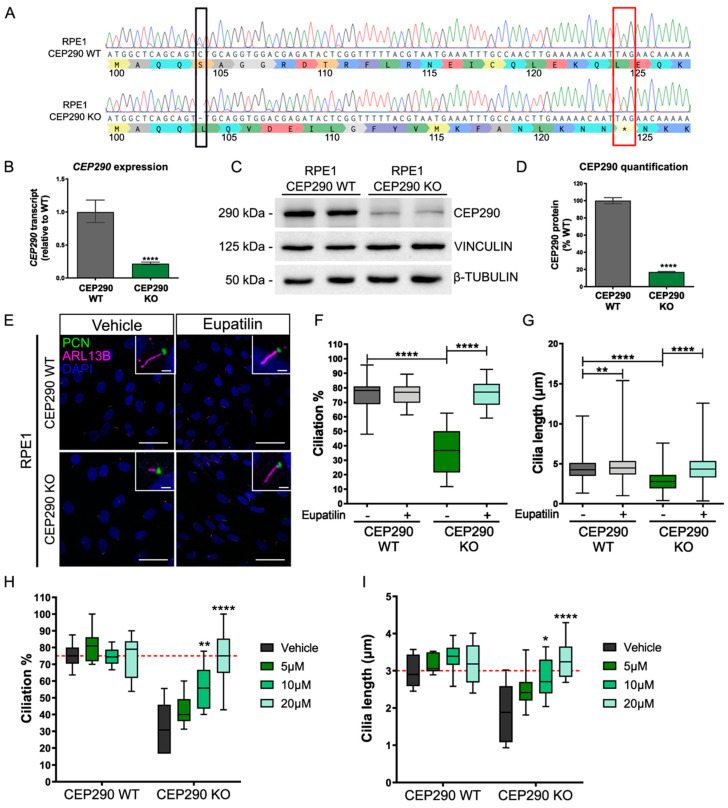
Eupatilin improves cilia incidence and length in CEP290 KO RPE1 cells. (**A**) Sanger sequence showing CRISPR/Cas9-induced deletion of 1 bp in *CEP290* (black box) and predicted effect on amino acid sequence. Stop codon is highlighted in exon 6 (red box). (**B**) qPCR analyses of *CEP290* transcripts show reduced levels in CEP290 KO RPE1 cells. Primers were designed between exon 13 and 14, *n* = 4. (**C**) Western blot of CEP290 protein shows reduced level in CEP290 KO cells, β-tubulin, and vinculin were used as reference proteins. (**D**) Quantification of Western blot, showing reduced CEP290 expression. *n* = 3 independent Western blots. (**E**) Representative images of CEP290 WT (control) and CEP290 KO RPE1 cells treated with vehicle or eupatilin (20 µM) with cilia stained for PCN (green) and ARL13b (magenta). Scale bar: 40 µm. Inset shows cilium at higher magnification. Scale bar: 2 µm. Graphical representation of cilia incidence (**F**), and cilia length (**G**), in WT and CEP290 KO RPE1 cells treated with 20 µM eupatilin. (**F**,**G**) 24–28 independent fields of view, from three separate experiment repeats were scored, and 207–630 cilia were measured. (**H**,**I**) The effect of eupatilin treatment for 24 h at the indicated doses on cilia incidence and length was measured. (**H**) eight independent images from two separate experiment repeats were scored, and 19–92 cilia were measured (**I**). Median ± min and max, * *p* < 0.05, ** *p* < 0.01, **** *p* < 0.0001 ordinary one-way ANOVA with Tukey’s post hoc test.

**Figure 3 cells-12-01575-f003:**
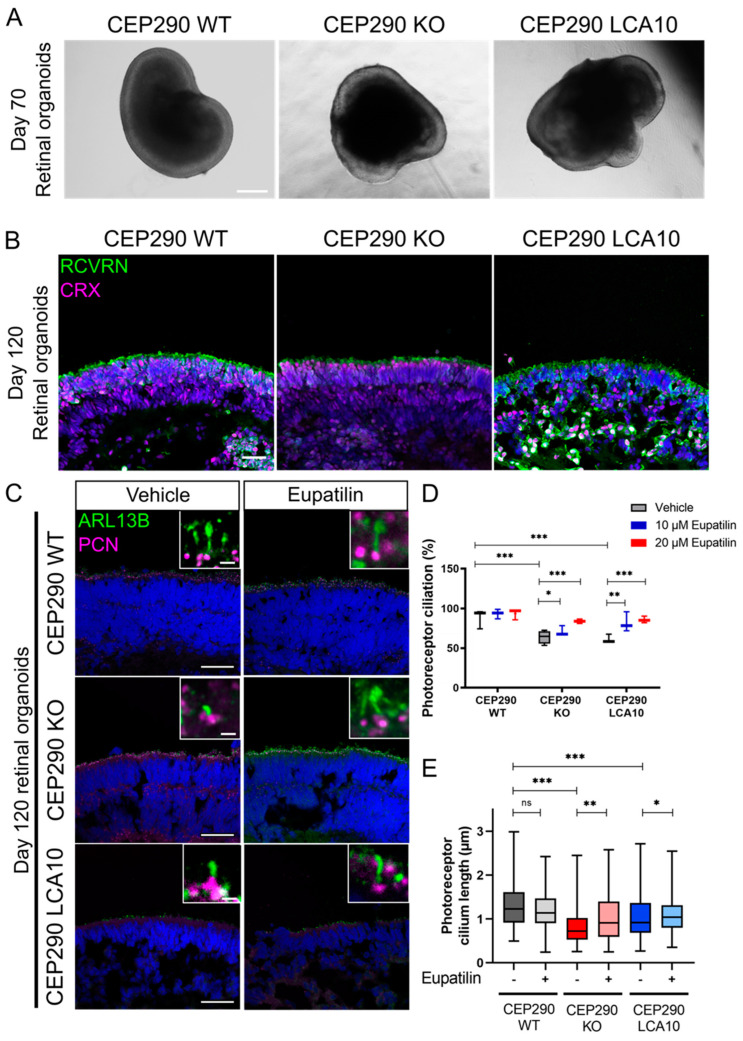
Eupatilin treatment restores ciliation and cilia length in iPSC-derived CEP290 KO and CEP290 LCA10 retinal organoids. (**A**) Bright field images of CEP290 retinal organoids at differentiation day 70. Scale bar: 500 µm. (**B**) RCVRN (green) and CRX (magenta) staining of day 120 retinal organoids. Scale bar: 40 µm. (**C**) Eupatilin treatment of retinal organoids from days 90 to 120. Cilia were stained with ARL13B (green) and PCN (magenta). Scale bar: 40 µm. Insets show a higher magnification of the photoreceptor cilia. Scale bar: 1 µm. (**D**) Quantification of photoreceptor ciliation and photoreceptor cilium length (**E**) in CEP290 retinal organoids at day 120 after eupatilin treatments. CEP290 WT DMSO *n* = 4 ROs (141 cilia), CEP290 WT EUP *n* = 4 ROs (135 cilia), CEP290 KO Vehicle *n* = 3 (148 cilia), CEP290 KO EUP *n* = 2 (88 cilia), CEP290 LCA10 Vehicle *n* = 4 (114 cilia), CEP290 LCA10 EUP *n* = 4 (176 cilia). * *p* < 0.05 ** *p* < 0.01, *** *p* < 0.001, ordinary one-way ANOVA with Tukey’s post hoc test.

**Figure 4 cells-12-01575-f004:**
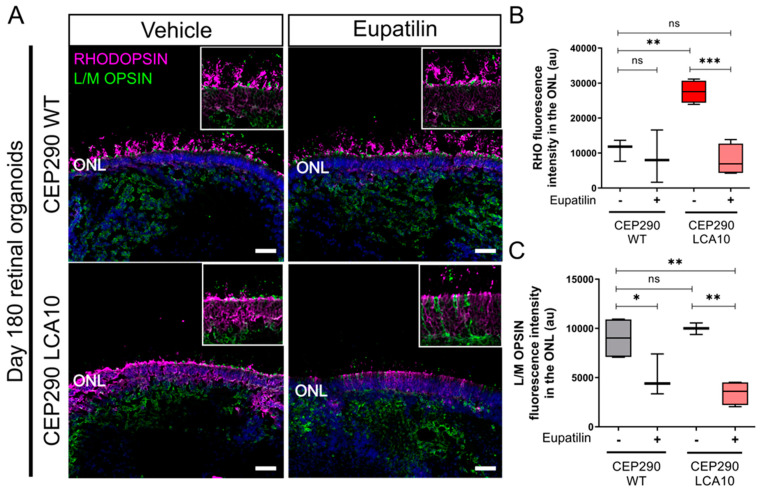
Opsin accumulates in the outer nuclear layer of CEP290 LCA10 retinal organoids and is rescued by eupatilin. (**A**) Representative images of CEP290 WT (control) and CEP290 LCA10 retinal organoids at D180 treated with eupatilin (20 µM) or vehicle for 30 days (from D150), stained for rhodopsin (magenta) and L/M opsin (green). Photoreceptor nuclei were stained with DAPI (blue). Scale bar: 10 µm. (**B**) Quantification of RHO and L/M opsin (**C**) immunofluorescence intensity in the photoreceptor outer nuclear layer (ONL). Each data point represents the mean of three measurements from one organoid, bar represents mean ± SD. * *p* < 0.05 ** *p* < 0.01, *** *p* < 0.001.

**Figure 5 cells-12-01575-f005:**
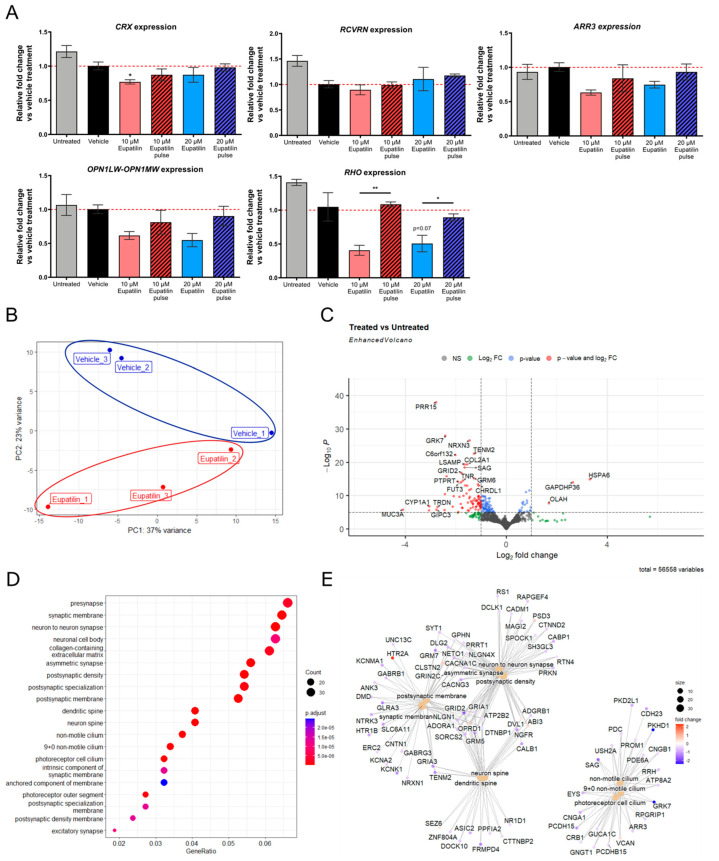
Transcriptome analysis of eupatilin treated control retinal organoids. (**A**) The effect of eupatilin on of *CRX*, *ARR3*, *RCVRN*, *RHO*, *OPN1LW-OPN1MW* gene expression by qPCR. Day 250 control retinal organoids were treated with vehicle (DMSO) or eupatilin at the indicated concentrations for 14 days and harvested or allowed to recover for 14 days with no treatment (“pulse”); untreated organoids are shown at D250 for comparison. qPCR analyses. Means ± SD *n* = 3 independent retinal organoids, * *p* < 0.05 ** *p* < 0.01, one-way ANOVA with Tukey’s post hoc test. Effect of eupatilin on the transcriptome by. Bulk RNA sequencing was performed on day 240 organoids after 1-month eupatilin treatment. (**B**), PCA analyses showing separation between the eupatilin treated organoids and the vehicle treated organoids by PC2 variance. (**C**), Volcano plot showing the most differentially expressed genes. (**D**), Dotplot showing the top 20 gene ontology (GO) cellular components by over-representation analysis. (**E**), Netplot showing the relationship between the top 10 most-enriched gene ontology terms and their genes.

## Data Availability

The data that support the findings of this study are available from the corresponding authors, J.C.C.-S. and M.E.C., upon reasonable request.

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
