# Peer review of "Eupatilin Improves Cilia Defects in Human CEP290 Ciliopathy Models"

_cells, 2023, doi:10.3390/cells12121575_

Round 1
Reviewer 1 Report
The manuscript « Eupatilin improves cilia defects in human CEP290 ciliopathy models » by Corral-Serrano et al. report the investigation of the effects of the flavonoid eupatilin on the improvement of cilium formation and length in four different models: LCA10 fibroblasts, CEP290 KO RPE1 cells, LCA10 and CEP290 KO iPSCs-derived retinal organoids.
They have showed the beneficial effect of eupatilin on different CEP290 ciliopathy models, including CEP290 retinal organoids. They also provide insights into the effect of eupatilin at the transcriptional level in control retinal organoids. Taken together, this work supports eupatilin as a potential variant-independent approach for CEP290-associated ciliopathies.
The paper is scientifically interesting and contains new knowledge and hope for therapeutic treatment for several ciliopathy.
I had just small questions or comments:
>Could you, please, add some information concerning your protocol for culture fibroblasts and RPE1 cells? You performed a starvation of 24 hours but we have no idea of when and how long you let the eupatilin?
>In the results: 3.2. Eupatilin restores cilia incidence and length in CEP290 knockout RPE1 cells
“Characterization of this cell line showed that the expression of CEP290 transcript was reduced by 80%, presumably due to nonsense mediated decay “
Could you see the band corresponding to the 20% of transcript by PCR? Is it possible to see two bands? One corresponding to transcript without exon 6 and the other to the mutated transcript (with only one base deleted).
Did you perform the sequencing of these transcripts?? (show it)
I can't see the supplementary data (the link www.mdpi.com/xxx/s1, Supplemental Material is not available). i will be interested by to see the imunolabelling.
Author Response
Please see attachment for the author's reply

Reviewer 2 Report
Review of Corral-Serrano et al.
Eupatilin improves cilia defects in human CEP290 ciliopathy models.
In this article, the authors investigate the ciliopathy non-syndromic Leber Congenital Amaurosis (LCA10), which is/can be caused by deep intronic mutations in the cilium-associated gene CEP290. The authors wanted to investigate a broader-spectrum approach than targeted gene editing/antisense corrections of the mutations that lead to the disease. They worked with the flavonoid eupatilin to treat various cell lines derived from either LCA10 mutation or CEP290 knockout cell lines.
The cell lines were 1) fibroblasts derived from LCA10 patients, 2) CEP290 KO RPE1 cell lines, and 3) retinal organoids derived from iPSCs with either LCA10 or knockout mutations in CEP290.
Their end outcomes measured were 1) cilium formation; 2) cilium length; AND most interestingly, 3) they demonstrated a functional outcome, which is reduced rhodopsin retention in the outer nuclear layer of LCA10 retinal organoids.
The authors start off with a dermal fibroblast cell line from an LCA10 patient and confirm previous research (Ref. Nr. 23: Parfitt et al. Cell Stem Cell 2016) showing restoration of both ciliation and cilia length, though with the latter measure the effect is strikingly small despite the very small p values (a result of very high n, presumably).
The authors then generate their own CEP290 knockout RPE1 cell lines and perform a very thorough analysis of this novel cell line. In fact, this figure (Nr. 2) represents the single most valuable scientific contribution of this research article. The dose-response curve for eupatilin is particularly convincing.
The authors then generate their own CEP290 knockout and CEP290 LCA 10 iPSC cell lines and differentiate the cells into retinal organoids. Again, they see a restoration of ciliation upon eupatilin treatment, with again a slight increase in cilium length as a result of eupatilin treatment.
The authors go on to use a very interesting functional assay to examine accumulation of rvisual opsins in the outer nuclear layer, which is presumably a sign of a defect in the ciliary transport machinery bringing the rhodopsin into the photoreceptor segment of the neurons (though the authors speculate that this may reflect changes in RHO expression levels, which also makes sense).
The authors conclude with some targeted qPCR analyses and whole-organoid RNAseq analysis, yielding potentially further new avenues for subsequent exploration.
Major point of critique:
All in all, this paper represents a huge amount of work, very carefully done, and the establishment of a fun new reagent for other researchers in the field. My biggest problem, and really the only problem that I have, with this paper is that their major finding has already been reported, as the authors note themselves (Ref. Nr. 17: Kim, Kim, et al., JCI 2018) and using a much more translationally-relevant benchmark, namely increased photopic response. A subsequent article (Ref. Nr. 18: Garcia-Gonzalo et al. MBC 2021) showed a rescue of ciliary gating in mouse embryonic fibroblasts. Have they contributed anything new to our understanding of the field, or is this article merely a technical advance carefully documenting the invention of a new cell line?
I am afraid I am not enough of an expert in the field to judge that for certain, so that will have to be the decision of the journal editors.
Minor points:
The manuscript is very well written, the figures are very nicely put together, and the methods section is clear and comprehensive.
This reviewer was not familiar with the concept of a "deep" intronic mutation, but a quick Internet search clarified that term. The authors may consider defining what this term means, as it would help to understand the genetic etiology of the disease.
Author Response
Please see the attachment for the author's reply

Reviewer 3 Report
In this paper, Corral-Serrano et al. present pre-clinical investigations into the flavonoid eupatilin as a potential treatment for retinal ciliopathies. Using four independent cellular models, including gene edited RPE and iPSC cells, as well as fibroblasts and retinal organoids derived from an LCA10 patient, the authors demonstrate that eupatilin treatment improves the ciliopathy phenotypes caused by CEP290 gene deficiency. The results of the study are compelling and significantly extend the findings of the previous report showing the beneficial effects on cilia development in CEP290 deficient RPE1 cells. The paper was very well written. Minor suggestions for improving the paper are provided below.
1. Figure 4: The results shown in figure 4 are drawn from a single organoid from each group - the presentation of technical variance in this figure is inconsistent with the biological variance presented in other graphs. The figure shows outer segment development in the control organoids, however it is unclear whether outer segments are present in the day 180 LCA10 retinal organoids. Outer segments are visible in brightfield micrographs – are any brightfield images available for the day 180 and day 250 retinal organoids? The description of these results focuses on rhodopsin immunoreactivity in the ONL - the authors should also comment on outer segment development/immunoreactivity observed in this experiment.
2. Line 83: Although a reference is provided, could the authors could provide some basic information about the patient and controls from whom the iPSC were derived? How were iPSC lines validated for pluripotency (karyotyping, gene expression, differentiation etc)?
3. Line 84: Ref 24 describes the generation of OPA1 iPSC, not CEP290 iPSC. The citation could be moved to the end of the sentence.
4. The Figures could be repositioned to appear after the first mention in the text.
Author Response

(The authors gave the same response as above.)

Round 2
Reviewer 1 Report
Dear authors
The article "Eupatilin improves cilia defects in human CEP290 ciliopathy mode" adds new scientific knowledge and hope for the treatment of ciliopathies. Comments from the review have been taken into account. I support the publication.
Reviewer 2 Report
I stand by my previous commentary, I don't think this work represents any novel advance, just a technical iteration. If the editors decide that technical reports have a place in this journal, so be it.